# The relationship between chronic health conditions and cognitive deficits in children, adolescents, and young adults with down syndrome: A systematic review

Kellen C. Gandy[1], Heidi A. Castillo[2], Lara Ouellette[3], Jonathan Castillo[2], Philip J. Lupo[1,4], Lisa M. Jacola[5], Karen R. Rabin[1,4], Kimberly P. Raghubar[6], Maria M. Gramatges[1,4]*

1 Department of Pediatrics, Section of Hematology/Oncology, Baylor College of Medicine, Houston, Texas, United States of America, 2 Department of Pediatrics, Section of Developmental Pediatrics, Baylor College of Medicine, Houston, Texas, United States of America, 3 Department of Research and Instruction, Texas Medical Center Library, Houston, Texas, United States of America, 4 Dan L. Duncan Cancer Center, Houston, Texas, United States of America, 5 Department of Psychology, St. Jude Children's Research Hospital, Memphis, Tennessee, United States of America, 6 Department of Pediatrics, Section of Psychology, Baylor College of Medicine, Houston, Texas, United States of America

* gramatge@bcm.edu

**Data Availability Statement:** All relevant data are within the manuscript and its Supporting Information files.

## Abstract

### Background

Individuals with Down syndrome are predisposed to a number of chronic health conditions, but the relationship between these conditions and cognitive ability is not clear. The primary objective of this systematic review is to assess this relationship by evaluating studies that measure cognitive performance in the context of Down syndrome-associated chronic health conditions.

### Methods

A systematic review was conducted in accordance with the Preferred Reporting Items for Systematic Review and Meta-Analysis (PRISMA) guidelines. Studies included in this review (1) included children, adolescent, and young adult participants with Down syndrome and one or more co-occurring health conditions; (2) were quantitative; and (3) reported outcomes related to both chronic health conditions and cognitive performance. A set of predetermined chronic health conditions that are common in Down syndrome (e.g. sleep disorders, congenital heart disease, thyroid disease, seizure disorders, and pulmonary hypertension) were selected based on prevalence rates in Down syndrome.

### Results

Fifteen studies met inclusion criteria. The majority these of studies assessed cognitive performance in association with sleep disorders (47%) and congenital heart disease (47%). Fewer studies reported on the effect of thyroid disease (7%) and seizure disorders (7%) on

**Funding:** This work was supported by a grant from the Jeffrey Pride Foundation to KRR, PJL, and MMG (no grant number specified). http://www.jeffreypridefoundation.org The funders had no role in study design, data collection and analysis, decision to publish, or preparation of the manuscript.

**Competing interests:** The authors have declared that no competing interests exist.

cognitive ability. None of the studies reported cognitive outcomes related to pulmonary hypertension. Of the chronic health conditions evaluated, associations between sleep disorders and cognitive dysfunction were most common among individuals with Down syndrome.

## Conclusions

Individuals with Down syndrome exhibit deficits in cognitive ability, particularly related to attention, executive function and verbal processing. These deficits may be further exacerbated by the presence of chronic health conditions, particularly sleep disorders. Individuals with Down syndrome and co-occurring sleep disorders may benefit from early interventions to mitigate their risk for adverse cognitive outcomes.

## Introduction

Individuals with Down syndrome (DS) exhibit impairments in cognitive ability, and are at increased risk for a wide range of chronic health conditions. Common comorbidities observed in DS include hearing or vision loss (60–75%), sleep disordered breathing such as obstructive sleep apnea (OSA) (70%), congenital heart defects (50%), thyroid disorders (4–18%), seizure disorders (1–13%), and persistent pulmonary hypertension (5%) [1–3]. In the general pediatric population, these conditions impact cognitive ability to varying degrees, contributing to deficits in attention, memory, executive function, visual-motor integration, and associated with lower intelligence quotient (IQ) [4–16]. While the above-described health conditions are common in DS, to date, only neurosensory deficits have been assessed for a relationship with cognitive ability [17, 18]. The objective of this systematic review is to address this gap in knowledge by determining from available evidence the impact of non-neurosensory chronic health conditions on cognitive functioning in DS.

Individuals with DS have a mild to moderate intellectual disability. The cognitive phenotype characteristic of DS is variable but includes deficits in attention, executive function (i.e., a broad range of higher-order skills that are necessary for goal directed behavior, including inhibition, working memory, cognitive flexibility, set-shifting, planning, and behavior regulation), expressive language, verbal processing, and explicit long-term memory, whereas certain aspects of visuospatial ability, associative learning, and implicit long-term memory may be relatively preserved [19]. Although cognitive deficits characteristic of DS often occur in the presence of multiple co-occurring health conditions [20], the extent to which specific conditions influence cognitive ability has not been systematically investigated. The objective of this systematic review is to evaluate and summarize evidence for a relationship between chronic health conditions commonly observed in DS and cognitive disability. Comparisons of cognitive ability (i.e. outcome data) were made between individuals with DS and diagnosed with a prespecified health condition (i.e. sleep disorders, cardiovascular disorders, thyroid disorder, seizure disorder, pulmonary disorder) and individuals with DS and without these health conditions.

## Materials and methods

A systematic review was conducted in collaboration with an academic medical librarian, and in accordance with the Preferred Reporting Items for Systematic Review and Meta-analyses (PRISMA) guidelines [21] (S1 Table), rather than from an existing protocol specific to this

investigation. Using the MEDLINE (PubMed, PsycINFO) and EMBASE databases, a search was performed for all English-language articles published between 1948 and October, 2019 using the following Medical Subject Heading (MESH) terms 'down syndrome', 'trisomy 21', 'mongolism' AND 'chronic health conditions', 'chronic diseases', 'comorbid*', 'multimorbid*', 'congenital heart disease', 'congenital heart malformation', 'heart defect', 'cardiovascular disorder', 'hypertension,' 'hyperthyroid', 'hypothyroid', 'thyroid disorder', 'thyroid disease', 'sleep apnea', 'sleep disturbances', 'sleep disorder*', 'sleep interrupt*', 'epilepsy', 'infantile spasms', 'seizures', 'cardiopulmonary', 'pulmonary disease', 'pulmonary disorder', 'pulmonary defect' AND 'memory', 'memories', 'attention', 'executive function', 'intelligence quotient', 'IQ', 'quotient*', 'intelligen*', 'cognitive', 'neurocognitive', 'neurodevelop*', 'cognition', 'visual-spatial', 'visual-integrat*', 'visual motor', 'motor skill', 'reaction*', 'latenc*', and 'processing speed' (completed: October 9, 2019). All studies yielded by this search strategy were then extracted to a database by the academic medical librarian.

All articles underwent two rounds of screening by two independent reviewers (KG, HC), in accordance with the Cochrane guidelines for the reporting of systematic reviews [22]. In the initial round, titles and abstracts were reviewed to determine whether articles met designated inclusion criteria: 1) study conducted in children, adolescents, and/or young adults with DS who were diagnosed with at least one co-occurring health condition (limited to sleep disordered breathing, cardiovascular disorder, thyroid disorder, seizure disorder, pulmonary disorder), 2) study reported outcomes for cognitive performance in the context of chronic health conditions, employing standardized metrics for assessing neurocognitive outcomes, i.e. formal assessments of attention, memory, and executive function. The definition of executive function employed for this review is a broad construct of higher-order processes that can be measured through performance-based cognitive testing or proxy-reported standardized ratings from the perspective of a caregiver, parent or teacher. Therefore, studies that employed informal observations or non-standardized metrics of cognition were excluded. Given that the focus of this review was on children, adolescents and young adults with DS, studies that included older participants with a comorbid diagnosis of Alzheimer's disease and/or dementia were also excluded. Case reports, review articles, validity/reliability studies, conference abstracts, editorials, and animal studies were excluded at this stage.

In the second round of review, both reviewers then conducted a full text screening of the refined list to determine the final number of included studies, ensuring that each paper met the designated inclusion criteria. Discrepancies between reviewers that were related to article eligibility were presented to the study team and a consensus was reached through discussion. To identify additional publications not discovered using traditional indexing resources, all articles referenced by the included studies also underwent full text review. The risk of bias was assessed for each included article using the Cochrane Risk of Bias in Systematic Reviews (ROBIS) tool (S2 Table) [23]. Risk of bias was assessed across four different domains related to 1) study eligibility criteria, 2) identification and selection of studies, 3) data collection and study appraisal, and 4) synthesis and findings. The risk of bias for each of the domains, as well as for the overall risk of bias for in the review, were rated as unclear, low, or high. Only studies with a low risk of bias were included in the systematic review.

## Results

We identified 7,990 articles, of which 34 met criteria for full text review and 15 were determined eligible for inclusion in this review (Fig 1). All of the studies included individuals with DS between the ages of 0 to 18 years, and in three studies extended to young adults up to 31 years. The included articles comprised 7 prospective cohort studies, 4 cross-sectional studies,

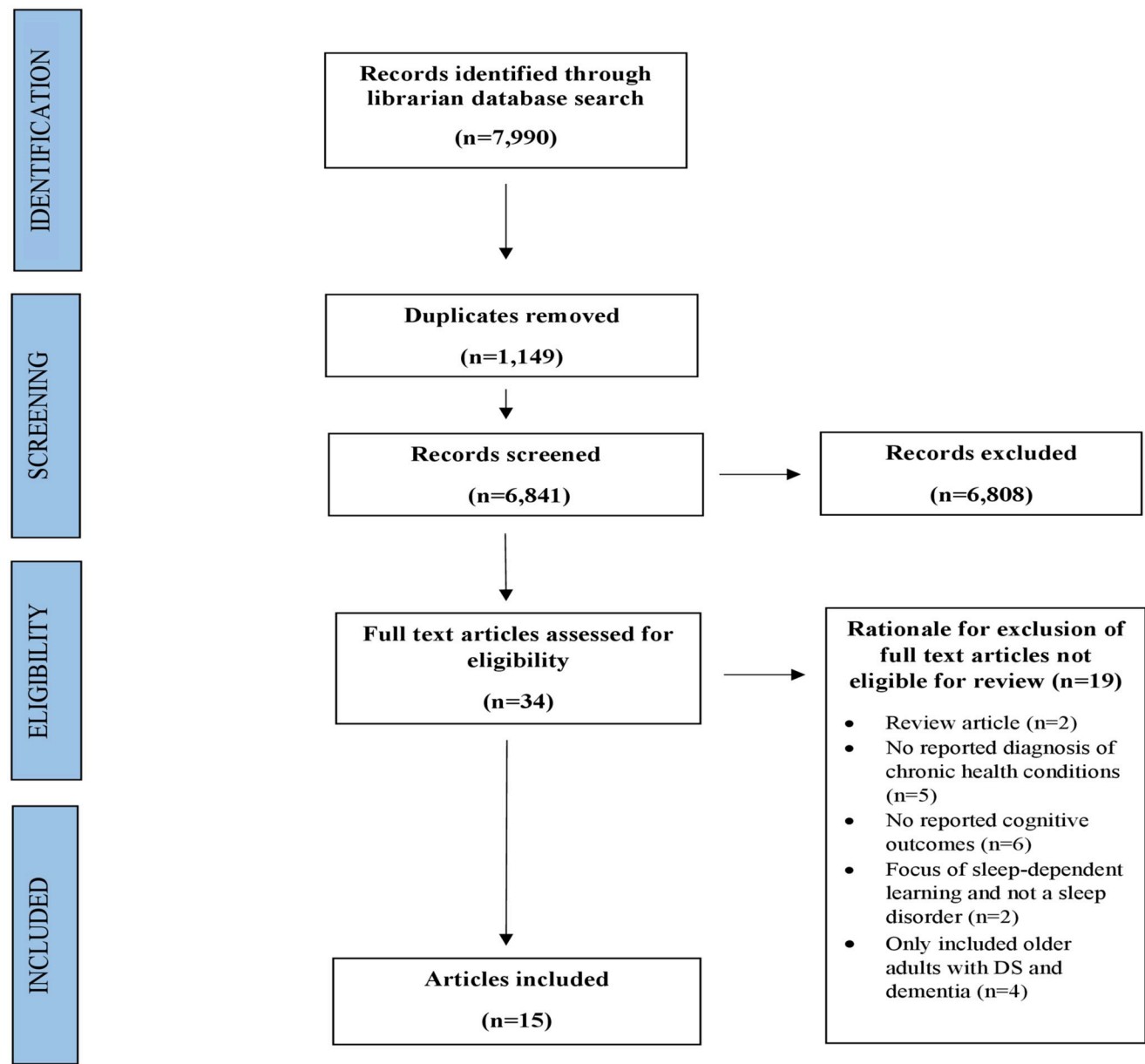

**Fig 1.**

and 4 case-control studies that were predominantly conducted in the U.S. and U.K., as well as in Israel, Greece, Thailand, and Taiwan. Sample sizes varied from 12 to 226 (mean 64, median 38). Included articles are summarized in Table 1, categorized by type of chronic health condition assessed. Risk of biases is shown in S2 Table (see **Supporting information**).

## Sleep disordered breathing and obstructive sleep apnea

Seven of fifteen articles (47%) reported on the spectrum of respiratory sleep disorders observed in association with cognitive performance in DS [24–31]. In six out of seven articles, presence of a respiratory sleep disorder was determined by polysomnography or an equivalent measure (domiciliary cardio-respiratory polygraphy, employed by Joyce, et al., [28] and [29]), and in

**Table 1. Summary of studies examining the associations between chronic health conditions and cognitive performance in individuals with DS.**

| Author(s), Year; CHC Category | Study Type (location) | Age range (years): n Population | Neurocognitive measure(s) | Neurocognitive Scoring/ Domains | Neurocognitive Results |
|---|---|---|---|---|---|
| **Andreou et al.**, 2002 Sleep disorder | Prospective cohort (Greece) | 17–25: 12 DS +SDB | Raven Progressive Matrices (RPM) | **Raw**: Visual-Spatial, Reasoning 1) Sum of correct items | SDB negatively correlated with visual-spatial scores, particularly in adults with DS+ untreated SDB. |
| **Chen et al.**, 2013 Sleep disorder | Prospective cohort (U.S.) | 14–31: 29 DS +parent- rated OSA | 1) Choice Reaction Time test 2) Verbal Fluency test 3) Knock-Tap Test | **Raw**: Executive Function 1) mean reaction time 2) sum of words generated 3) sum of correct responses | The severity of parent-rated OSA in DS moderately correlated with executive function performance. More severe symptoms predicted poorer performance. |
| **Breslin et al.**, 2014 Sleep disorder | Prospective cohort (U.S.) | 7–12: 38 DS ±OSA | Cambridge Neuropsych. Test Automated Battery (CANTAB): 1) Intra-Extra Dimensional Set Shifting 2) Paired-Associative Learning Task 3) Simple Reaction Time Task Kaufman Brief Intelligence Test (KBIT-II) | **Raw**: Executive Function 1) Stages completed 2) Mean errors 3) Median reaction time **SS**: Full, Verbal, Non-verbal IQ | DS+OSA completed fewer stages for the set shifting task of cognitive flexibility. Reaction time and mean errors did not differ between groups. DS+OSA had lower verbal IQ. No differences in full-scale or non-verbal IQ. |
| **Brooks et al.**, 2015 Sleep disorder | Prospective cohort (U.S.) | 6–14: 25 DS ±OSA | Stanford-Binet Intelligence Scales, 4th Edition (SB-4) Beery-Buktenica Developmental Test of Visual-Motor Integration (Berry VMI) Conners Hyperactivity Index *Parent rept | **SS**: Full IQ, Bead Memory, Vocabulary, Quantitative Pattern Analysis, Memory for Sentences, Comprehension **SS**: Visual, Motor, and Visual-Motor Integration **SS**: Attention | No differences in cognitive performance between DS children with and without OSA. Total sleep time and latency contributed to cognitive performance. Of 10 DS+OSA, the 5 treated with adenotonsillectomy and/or CPAP showed improvements in parent-reported attention. |
| **Joyce et al.**, 2017 Sleep disorder | Case control (U.K.) | 2–5: 44 DS +SDB: 22 TD +SDB: 22 | Mullen Scales of Early Learning (MSEL) | **Sc**: Gross motor, fine motor, visual reception, receptive language, expressive language **Raw**: sum of total score | SDB in preschoolers with DS not associated with cognitive outcomes involving motor skills and language. |
| **Joyce et al.**, 2019 Sleep disorder | Prospective cohort (U.K.) | 3–6: 80 DS ±OSA | Behavior Rating Inventory of Executive Function–Preschool Version (BRIEF-P) *Parent rept | **T score**: Inhibit, shift, emotional control, working memory, inhibitory self-control index, flexibility index, emergent metacognition index, global executive composite, plan/organize | OSA associated with poorer working memory, emotional control, and shifting (aspects of executive function). |
| **Lee et al.**, 2019 Sleep disorder | Prospective cohort (Taiwan) | 6–18: 30 DS ±OSA | Wechsler Preschool and Primary Scale of Intelligence (WPPSI-R) Wechsler Intelligence Scale for Children (WISC-IV) Wechsler Adult Intelligence Scale (WAIS-III) Developmental Neuropsychological Assessment (NEPSY) visuomotor precision | **Sc**: Vocabulary (this was the only subtest reported in the results) **SS**: FSIQ **Sc**: Fine motor skills | After adjustment for age and FSIQ, DS+OSA not associated with deficits in vocabulary or visuomotor precision. In a subgroup of ages 6–12 years, OSA associated with lower WPPSI-R vocabulary. No associations with visuomotor precision. |
| **Rihtman et al.**, 2010 Cardiovascular disorder | Cross-sectional (Israel) | 6–17: 60 DS ±other congenital anomalies (36 CHD) | Stanford-Binet Intelligence Scales (SB-4) Beery-Buktenica Developmental Test of Visual-Motor Integration (Berry VMI) | **SS**: Full IQ, Verbal, Abstract/Visual, Qualitative Reasoning, Short-Term Memory **SS**: Visual, Motor, and Visual-Motor Integration | Presence or severity of congenital anomaly (i.e. CHD) not associated with cognitive performance. |
| **Visootsak et al.**, 2011 Cardiovascular disorder | Case control (U.S.) | 0.5–2: 29 DS +CHD: 12 DS-CHD: 17 | Bayley Scales of Infant and Toddler Development (Bayley-III) | **SS**: Cognitive, Language, and Motor. | Infants with DS+CHD had lower motor composite scores, but no differences in cognitive or language composite scores. |

*(Continued)*

Table 1. (Continued)

| Author(s), Year; CHC Category | Study Type (location) | Age range (years): n Population | Neurocognitive measure(s) | Neurocognitive Scoring/ Domains | Neurocognitive Results |
|---|---|---|---|---|---|
| **Visootsak et al.**, **2013** Cardiovascular disorder | Case control (U.S.) | 2–3.5: 29 DS +CHD: 12 DS-CHD: 17 | Mullen Scales of Early Learning (MSEL) | **Sc**: Expressive Language, Receptive Language, Visual Reception, Fine Motor | No differences in expressive, receptive language, or visual and fine motor. |
| **Alsaied et al.**, **2016** Cardiovascular disorder | Cross-sectional (U.S.) | 0–2: 36 DS +CHD: 12 DS-CHD: 24 3–5: 38 DS+CHD: 7 DS-CHD: 31 6–18: 104 DS+CHD: 26 DS-CHD: 78 | Bayley Scales of Infant and Toddler Development (Bayley-III) Peabody Developmental Motor Scales (PDMS-II) Brief Rating Inventory of Executive Function (BRIEF) *Parent rept Differential Ability Scales (DAS-II) Stanford-Binet Intelligence Scales (SB-5) | **SS**: Language, Motor **Sc**:" Receptive Language, Expressive Language, Fine Motor, Gross Motor **Raw**: Visual Motor, Grasping **SS**: Fine Motor **T score**: Executive function, Initiate, Global executive composite, Behav. Regulation, Emotional control, Inhibit, Plan/ Organize, Monitor, Working memory, Composite organization of materials **SS**: Nonverbal IQ **SS**: Full IQ, Knowledge, Qualitative Reasoning, Visual Spatial | Infants with DS+CHD had lower receptive, expressive, and language scores. No differences in fine/gross motor scores. Preschoolers with DS+CHD did not differ in auditory or expressive language, visual motor, grasping, or fine motor scores. Children with DS+CHD did not differ in executive function, nor in performance-based measures of nonverbal IQ, full IQ, knowledge, qualitative reasoning, and visual spatial skills. |
| **Visootsak et al.**, **2016** Cardiovascular disorder | Case-control (U.S.) | 0.5–2: 57 DS +CHD: 20 DS-CHD: 37 | Bayley Scales of Infant and Toddler Development (Bayley-III) | **Raw Composite**: Cognitive, Expressive Language, Receptive Language, Gross Motor, Fine Motor | In adjusted analyses, DS+ASVD had lower gross motor and cognition/ expressive language scores. |
| **Rosser et al.**, **2018** Cardiovascular disorder (among other CHC) | Cross-sectional (U.S.) | 6–25: 226 DS +CHD:122 DS-CHD: 104 | Arizona State Cognitive Battery: CANTAB, reaction time, paired-associate learning, spatial span, intra/extradimensional set-shifting Virtual computer-generated arena Modified dots task Finger sequencing task Developmental Neuropsychological Assessment (NEPSY) visuomotor precision Kaufman Brief Intelligence Test (KBIT-II) Scales of Independent Behavior-Revised (SIB-R)*Parent rept Behavior Rating Inventory of Executive Function-School Age (BRIEF)* Parent rept Nisonger Child Behavior Rating Form (NCBRF) *Parent rept | **Z scores**: Associative learning, reaction time, memory span, set-shifting **Raw**: Spatial learning **Raw**: Inhibitory control, working memory **Raw**: Sensorimotor function **Sc**: Fine motor skills **SS**: Full, Verbal, Non-verbal IQ **Sc**: Motor, social and communication, personal living, community living skills **T score**: Executive function, Initiate, Global executive composite, Behav. Regulation, Emotional control, Inhibit, Plan/ Organize, Monitor, Working memory, Composite organization of materials **Sc**: Compliant/calm, adaptive social, conduct problems, insecure/anxious, hyperactive, self-injury/stereotypic, self-isolated/ritualistic, overly sensitive | CHD requiring surgery in the first years of life was not associated with poorer cognitive ability for both performance-based and proxy-reported assessments, after adjusting for gender, race/ethnicity, socioeconomic status. |
| **Wasant et al.**, **2008** Cardiovascular & thyroid disorders (among other CHC) | Cross-sectional (Thailand) | 3–6: 100 | Capute Scales of Cognitive Adaptive Test/Clinical Linguistic and auditory Milestones Scales (CAT/CLAMS) | **SS**: Developmental Quotient (DQ) | CHD, but not thyroid disorders, was associated with lower DQ. Family income and age at the first speech-training program were the only independent predictors of DQ. |

(*Continued*)

**Table 1.** (Continued)

| Author(s), Year; CHC Category | Study Type (location) | Age range (years): n Population | Neurocognitive measure(s) | Neurocognitive Scoring/ Domains | Neurocognitive Results |
|---|---|---|---|---|---|
| **Tapp et al., 2005** Seizure disorder | Prospective cohort (U.S.) | 1–3: 29 DS+IS: 8 DS-seizures: 21 | Bayley Scales of Infant and Toddler Development (Bayley-III) | **Sc:** Receptive Language, Expressive Language, Fine Motor, Gross Motor **SS:** Cognitive | DS+IS scored lower across all cognition domains. IS treatment delays did not contribute to neurodevelopmental outcomes. |

DS = Down syndrome, TD = typically developing without Down syndrome, SDB = sleep disordered breathing, WS = William's syndrome, ID = Intellectual disability, CHC = Chronic health condition, CHD = Congenital heart disease, ASVD = Atrioventricular septal defect, ASD = Atrial septal defect, OSA = Obstructive sleep apnea, IS = infantile spasms, SQ = sleep quality, IQ = intelligence quotient, DQ = developmental quotient, MA = mental age, SS = standard composite score, SD = 15. Sc = Scaled scores based on age-normed T scores.

one article, OSA was determined by parent report (Sleep Questionnaire, Chen, et al.). Two articles reported on sleep disordered breathing (SDB) and five on obstructive sleep apnea (OSA). SDB was defined by the number of apnea events per hour, heart rate, and oxygen saturation, whereas OSA determined by polysomnography was defined by stricter criteria, specifically the number of apnea and hypopnea events per hour of sleep. The studies that examined effects of OSA on cognition used thresholds commonly employed in pediatric populations and ranging from $\geq 1$ - $\geq 1.5$ events/hour, although one study used an adult-defined cutoff of $\geq 5$ events/hour (Brooks, et al.) All seven studies employed a diverse set of performance-based cognitive assessments, noted for each study to measure aspects of executive function, visual-spatial skills, reasoning, attention, memory, motor skills, verbal fluency, cognitive flexibility and intelligence quotients (IQ).

Two articles reported on sleep disordered breathing (SDB) and seven on obstructive sleep apnea (OSA). In five of these studies, presence of OSA or SDB was associated with impaired visuospatial abilities, as well as deficits in aspects of executive function including slower reaction times, decreased verbal fluency, decreased cognitive flexibility as measured by set shifting ability, poorer emotional control, poorer working memory, poorer language function, and significantly lower verbal IQ [24, 26, 27, 29, 30]. One study reported on a potential age-effect of sleep disorders, showing that younger children with DS and OSA (ages 6–12 years) were more likely to have impaired expressive vocabulary as measured by a subtest of the Wechsler Preschool and Primary Scale of Intelligence, when compared with adolescents and young adults with DS between the ages of 12–25 years [30]. Frequency of sleep apnea per hour was also negatively correlated with visual-perceptual ability, as measured by the Raven's Progressive Matrices [32] in adolescents and young adults with DS, with increased apneic frequency associated with worse performance [24].

Three studies supplemented polysomnography with parent-reported sleep behaviors [26, 27, 31]. Breslin et al. found no difference in daytime sleepiness or total sleep time between DS children with and without OSA, noting poor correlation between parent-reported outcomes and polysomnography. Brooks et al. noted a relationship between sleep latency (time to sleep onset) and intelligence assessed by the Stanford-Binet. Chen et al. noted that difficulties with initiating and maintaining sleep was negatively correlated with verbal fluency performance. Two studies found no associations with sleep disorders and cognitive function [28, 31]; however, both employed non-standard definitions of OSA and SBD (see Discussion). In sum, the majority of studies that examined the effect of OSA and SBD in DS observed an increased risk for cognitive impairment in younger children and adolescents with DS and co-occurring sleep disorders.

## Cardiovascular disorders

Seven out of fifteen articles (47%) reported on cardiovascular disorders in association with cognitive performance, all of which focused on congenital heart disease (CHD) [33–39]. Again, a wide range of performance-based cognitive assessments were used to measure IQ, developmental quotients (DQ), visual-motor abilities, gross/fine motor skills, language, behavior, cognition, learning and memory, and aspects of executive function including set-shifting, behavioral and emotional control, working memory, and planning/organizing skills.

The most commonly reported disorder was congenital atrioventricular septal defect (AVSD), a moderately complex CHD often corrected within the first year of life. Among infants and toddlers between the ages of 6–36 months, there were no clear associations between CHD and cognitive ability, with mixed reports regarding composite scores of cognition, motor, and language revealing a significant association in some reports and no association in other reports [33–36, 38]. Studies that detected cognitive impairment assessed younger children with DS within three years of a surgical repair that had been performed during the first year of life [33–35]. In contrast, studies conducted in older children and young adults observed no impact of CHD on IQ, motor skills, language ability, spatial learning, sensorimotor function, and performance-based and parent-reported aspects of executive function [33, 37, 39]. In the largest of these cohorts (n = 234), there was no association between CHD and either performance-based or parental ratings of cognitive performance in children ages 6–25 years [39], suggesting that corrected CHD does not have a significant impact on long-term cognitive development. Although lower DQ was reported in one study that included children three to six years old [38], cognitive deficits were not noted among children with DS and CHD age ≥3 years, albeit these conclusions are limited by the cross-sectional nature of these studies.

## Thyroid disorders

One out of fifteen articles (7%) reported on thyroid disorders in association with cognitive performance in DS, together with multiple other co-occurring chronic health conditions. The study employed a performance-based cognitive assessment (Capute Scales of Cognitive Adaptive Test/Clinical Linguistic and auditory Milestones Scales) to measure DQ in children ages 3–6 years [40], and did not observe an association between DQ and early thyroid dysfunction, specifically, congenital hypothyroidism that was treated within the first 2 months of life [38].

## Seizure disorders

One out of fifteen articles (7%) reported on infantile spasms, a severe, refractory type of seizure disorder, in association with cognitive performance in DS [41]. The study employed a performance-based measure of IQ (Bayley Scales of Infant Development), which includes composite scores of cognition, language, and motor ability [42]. In comparison with age-matched children with DS and without a history of infantile spasms (1 to 3 years), children with DS and infantile spasms demonstrated significantly lower scores in domains of cognition, receptive language, expressive language, fine motor, and gross motor skills (i.e., 1.5 standard deviations lower across all domains) [41].

## Pulmonary disorders

None of the studies reported on pulmonary disorders and cognitive performance in DS, possibly reflecting a relatively low prevalence of pulmonary hypertension in children with DS.

## Discussion

In sum, evidence obtained in this systematic review suggests that respiratory sleep disorders, specifically SBD and OSA, are associated with poorer cognitive function in children and young adults with DS, particularly in the domains of verbal IQ, visual-perceptual skills and aspects of executive function including set shifting ability, emotional control, verbal fluency and working memory. OSA has been defined in pediatric populations as at least 1 apnea or hypopnea event per hour of sleep [43, 44]. The three studies that found an association between OSA and cognitive function adhered to this definition, whereas the two that found no association did not: one employed adult age-based thresholds for defining OSA ($\geq$5 events per hour of sleep) and the other used median cutoffs for defining high or low apnea hypopnea index in the study population, which may explain the lack of associations observed. When defined by polysomnography and using the above threshold, the incidence of OSA in DS participants for the included studies ranged from 51–80%, consistent with recent reports [3]. CHD was also associated with cognitive performance, but the results were less consistent and the effect limited to younger children with recent history of surgical repair, suggesting short-term rather than permanent impact on cognitive ability. Of note, although both CHD and SDB/OSA are common in DS, none of the included studies considered the contributions of these conditions, if co-occurring, on cognitive function. Additional studies are needed to address this deficit in our understanding of the impact of DS-associated chronic health conditions on cognitive function.

There was only one report of association between infantile spasms and cognitive performance, but the results were consistent with the higher rates of cognitive disability observed in conjunction in non-DS children diagnosed with this seizure disorder [45]. Lastly, although chronic, uncontrolled hypothyroidism is known to adversely impact cognitive ability [46], if corrected early there was no discernable impact on cognitive ability in children with DS. Chronic uncorrected hypothyroidism is rare, so that there are limited opportunities to assess the impact of untreated hypothyroidism on cognitive ability in DS.

In our review, poor quality sleep had the strongest and most consistent association with an adverse impact on cognitive development in children with DS, particularly in the areas of verbal IQ and aspects of executive function. The American Academy of Pediatrics guidelines for health supervision in children with DS include recommendations for obtaining a pediatric sleep study or polysomnogram prior to the age of four [1]. Early diagnosis and intervention may mitigate the detrimental effects of poor sleep quality during early brain development, a critical period of growth and high neuroplasticity, so that individuals with DS may reach their full cognitive potential.

Interventions that improve sleep quality are available, and can lead to subsequent improvements in cognitive ability. Early intervention should be considered to mitigate the negative effects of recurrent, long-term apnea events. Tonsillectomy and/or adenoidectomy (TA) is one of the more commonly used and successful surgical options available for treating childhood apnea [47]. In children with DS, TA improves sleep quality and reduces symptoms of OSA and reduces symptoms of OSA [48–51], suggesting potential benefit for mitigating the negative impact of OSA on cognitive ability. In severe cases, alternate and complementary surgical approaches may be employed [52, 53]. However, there is evidence for somewhat reduced efficacy of TA in mitigating OSA in children with DS, who often experience persistent moderate to severe residual OSA [49, 50]. Also of note, while early TA intervention improves nonverbal reasoning, fine motor skills, and selective attention in non-DS children [54], the impact of TA on cognitive outcomes in DS has not yet been investigated.

In cases where surgical intervention is not successful, continuous positive airway pressure (CPAP) device may also alleviate OSA symptoms. However, compliance with these devices is often poor, especially among individuals with DS [55]. While no DS-specific studies have been conducted to improve CPAP adherence through systematic desensitization or exposure techniques, such techniques have been successfully employed in DS to treat phobias [56, 57], and may also serve useful for increasing CPAP compliance. Use of devices designed for pediatric populations that feature brightly-colored fabric patterns and cushioned shapes may also reduce discomfort and encourage use.

Limitations to this review include a search strategy that was restricted to two databases and only English language articles. Only a small number of published studies were found that evaluated both cognitive performance and chronic health conditions in DS. In addition, the inconsistent methods employed for assessing cognitive ability precluded conducting a meta-analysis of the studies reviewed, necessitating a systematic review that was largely qualitative. Although there were associations noted between the above described chronic health conditions and cognitive performance in DS, it is important to note that the sample sizes for many these studies were relatively small, ranging from 12 to 69 participants for the majority of the studies (80%), and therefore may have lacked sufficient power to draw strong conclusions. Moreover, the lack of association between chronic health conditions and cognitive ability that we observed among older DS individuals may be related to a widening gap in cognitive ability between chronologically age-similar DS and non-DS children as they age, which can strengthen the floor effect over time. Although we included only those studies with a low risk of global bias (based on the Cochrane ROBIS guidelines), we identified independent potential biases to our review, such as studies with insufficient power to detect large effects due to a small sample size and a lack of reported sensitivity analyses (S2 Table), findings that were expected given the scarcity of literature on cognition related to health conditions in DS.

In sum, the impact of chronic health conditions on cognitive ability in DS is understudied. This systematic review is the first to assess the potential burden of chronic health conditions on cognitive performance in a neurodevelopmentally vulnerable population. Our findings suggest a need for more comprehensive characterization of the burden of chronic health conditions experienced by DS populations, and how these conditions relate to social, physical and cognitive ability. Further, the observed impact of respiratory sleep disorders on cognition warrants further study of sustainable, early interventions for children with DS that may mitigate the impact of this condition on cognitive potential.

## Supporting information

**S1 Table. PRISMA 2009 checklist.**
(DOC)

**S2 Table.**
(DOCX)

## Author Contributions

**Conceptualization:** Kellen C. Gandy, Philip J. Lupo, Karen R. Rabin, Kimberly P. Raghubar, Maria M. Gramatges.

**Data curation:** Kellen C. Gandy, Heidi A. Castillo, Lara Ouellette.

**Formal analysis:** Kellen C. Gandy, Heidi A. Castillo, Lara Ouellette.

**Funding acquisition:** Philip J. Lupo, Karen R. Rabin, Maria M. Gramatges.

**Methodology:** Lara Ouellette, Kimberly P. Raghubar, Maria M. Gramatges.

**Project administration:** Philip J. Lupo, Lisa M. Jacola, Karen R. Rabin, Kimberly P. Raghubar, Maria M. Gramatges.

**Resources:** Lara Ouellette, Jonathan Castillo.

**Supervision:** Jonathan Castillo, Philip J. Lupo, Lisa M. Jacola, Karen R. Rabin, Kimberly P. Raghubar, Maria M. Gramatges.

**Validation:** Kimberly P. Raghubar, Maria M. Gramatges.

**Writing – original draft:** Kellen C. Gandy.

**Writing – review & editing:** Heidi A. Castillo, Jonathan Castillo, Philip J. Lupo, Lisa M. Jacola, Karen R. Rabin, Kimberly P. Raghubar, Maria M. Gramatges.

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
