## [Decision Letter · Decision Letter 0]

26 Jun 2020

PONE-D-20-14548

The Relationship between Chronic Health Conditions and Cognitive Deficits in Children, Adolescents, and Young Adults with Down Syndrome: A Systematic Review

PLOS ONE

Dear Dr. Gramatges,

Thank you for submitting your manuscript to PLOS ONE. After careful consideration by 2 Reviewers and an Academic Editor, all of the critiques of both Reviewers must be addressed in detail in a revision to determine publication status. If you are prepared to undertake the work required, I would be pleased to reconsider my decision, but revision of the original submission without directly addressing the critiques of the two Reviewers does not guarantee acceptance for publication in PLOS ONE. If the authors do not feel that the queries can be addressed, please consider submitting to another publication medium. A revised submission will be sent out for re-review. The authors are urged to have the manuscript given a hard copyedit for syntax and grammar.

**Comments to the Author**

1. Is the manuscript technically sound, and do the data support the conclusions?

Reviewer #1: Partly

Reviewer #2: Partly

2. Has the statistical analysis been performed appropriately and rigorously? 

Reviewer #1: N/A

Reviewer #2: N/A

3. Have the authors made all data underlying the findings in their manuscript fully available?

Reviewer #1: Yes

Reviewer #2: Yes

4. Is the manuscript presented in an intelligible fashion and written in standard English?

Reviewer #1: Yes

Reviewer #2: Yes

5. Review Comments to the Author

Reviewer #1: This systematic review found 15 studies which investigated the relationship between chronic health conditions and cognition in children with Down syndrome (DS). The study used the gold standard of PRISMA for selecting the papers included in the systematic review. Almost half of the studies focused on sleep disorders which are very common in DS but a large number also focused on cardiovascular disorders particularly congenital heart disease which is also very common in the DS population. The summary of the studies appears in the supplementary Table S1 and I would urge the authors to include this table in the main body of the manuscript. The manuscript summarises important information regarding cognitive outcomes in children with DS and how these might be improved by treating other common chronic health conditions. However there are a number of limitations in the format in which the review is currently presented.

Major comments:

As sleep disorders and congenital heart disease are both extremely common in DS it is not surprising that most of the studies have related these disorders to cognitive function. What the review does not make clear is how these conditions were treated when they presented in the same child. Children with DS also are much more likely to be affected by sleep problems associated with the initiation and maintenance of sleep – and these are not mentioned see Rosen D, Lombardo A, Skotko B, Davidson EJ. Parental perceptions of sleep disturbances and sleep-disordered breathing in children with Down syndrome. Clin Pediatr (Phila). 2011;50:121-5; Hoffmire CA, Magyar CI, Connolly HV, Fernandez ID, van Wijngaarden E. High prevalence of sleep disorders and associated comorbidities in a community sample of children with Down syndrome. J Clin Sleep Med. 2014;10:411-9; Churchill SS, Kieckhefer GM, Bjornson KF, Herting JR. Relationship between sleep disturbance and functional outcomes in daily life habits of children with Down syndrome. Sleep. 2015;38:61-71. Up to 97% of children with DS also have obstructive sleep apnoea depending on the patient selection criteria, definitions and methodology used (Walter et al., Insights into the effects of sleep disordered breathing on the brain in infants and children: imaging and cerebral oxygenation measurements. Sleep Medicine Reviews. 2019 Dec 16;50:101251. doi: 10.1016/j.smrv.2019.101251). The review needs to discuss how sleep disorders in general and congenital heart disease may interact and it the papers identified if subjects had both sleep disorders and congenital heart disease.

Sleep disordered breathing (SDB) is the global term generally used to cover both central and obstructive breathing patterns during sleep. Obstructive SDB describes a spectrum of severity of disorders which ranges from primary snoring which is not associated with sleep disruption or gas exchange abnormalities to obstructive sleep apnoea (OSA). In children, OSA is usually defined as an obstructive apnoea hypopnoea index of > 1 event per hour of sleep. The authors need to rewrite the section of discussion which does not make it plain if the papers are talking about polysomnographically defined OSA or parental reports of SDB.

The section on treatment of OSA also needs to be rewritten as adenotonsillectomy is the first line of treatment for OSA in both typically developing children and children with DS. Adenotonsillectomy is less effective in children with DS as the authors report, and CPAP is frequently used when OSA is not completely resolved with adenotonsillectomy. Other forms of surgery are also common in children with DS include tracheostomy, mandibular distraction osteogenesis for children with significant retrognathia, genioglossus advancement, rapid maxillary advancement, lingual tonsillectomy, tongue reduction, tongue hyoid advancement or suspension, uvulopalatopharyngoplasty, tonsillar pillar plication, and laryngotracheoplasty (Rosen D. Management of obstructive sleep apnea associated with Down syndrome and other craniofacial dysmorphologies. Current Opinion in Pulmonary Medicine. 2011;17:431-6) and recently hypoglossal nerve stimulation (Diercks GR, Wentland C, Keamy D, Kinane TB, Skotko B, de Guzman V, et al. Hypoglossal Nerve Stimulation in Adolescents With Down Syndrome and Obstructive Sleep Apnea. JAMA Otolaryngol Head Neck Surg. 2017).

The co-occurrence of sleep disorders and congenital heart disease should also be discussed regarding thyroid and seizure disorders.

Reviewer #2: Thank you for giving me the opportunity to read this interesting systematic review on the relationship between cognitive deficits and chronic health conditions in people with Down Syndrome. The manuscript requires significant changes:

1.It is not clear why certain chronic health conditions were selected, e.g. why was hearing impairment (a common health condition) not included?

2. Was Mosaic Down syndrome included as it is not described as a search term.

3. Was hand searching and reference lists for articles carried out? if not, why was this not included as part of the methodology as it is standard practice. Please also state the date range within which articles were searched.

4. What tool was used to assess the risk of bias? please provide a reference.

5.How was data extracted? was this done by one or more raters?

6. No information is given on how the data was analysed. Given that quantitative outcomes were included, there should be an explicit statement of why a meta-analysis was not performed. There appears to be a narrative synthesis of data but there is no attempt to provide statistics for any of the relationships discussed.

7. In the results section, it would be helpful to provide a summary of how many studies were of each type, the variation in terms of sample sizes and where the studies were conducted (this information should be included in the tables)

8. In the discussion section, please provide a discussion of the limitations of the study (e.g. only three electronic databases searched, only English language articles etc...).

6. PLOS authors have the option to publish the peer review history of their article (what does this mean?). If published, this will include your full peer review and any attached files.

**Do you want your identity to be public for this peer review?** For information about this choice, including consent withdrawal, please see our Privacy Policy.

Reviewer #1: No

Reviewer #2: No

We look forward to receiving your revised manuscript.

Kind regards,

Stephen D. Ginsberg, Ph.D.

Section Editor

PLOS ONE

2. In your methods section, please state which validated tool was used to assess the risk of bias in the individual studies. For example, the Newcastle-Ottawa scale for observational studies.

---

## [Author Response · Author response to Decision Letter 0]

5 Aug 2020

Review Comments to the Author

Reviewer 1

Reviewer 1 requested that we include Table S1, a summary of the studies reviewed, in the main body of the manuscript. This has been done, and is now shown as Table 1. 

1. As sleep disorders and congenital heart disease are both extremely common in DS it is not surprising that most of the studies have related these disorders to cognitive function. What the review does not make clear is how these conditions were treated when they presented in the same child. Children with DS also are much more likely to be affected by sleep problems associated with the initiation and maintenance of sleep – and these are not mentioned.

see Rosen D, Lombardo A, Skotko B, Davidson EJ. Parental perceptions of sleep disturbances and sleep-disordered breathing in children with Down syndrome. Clin Pediatr (Phila). 2011;50:121-5; Hoffmire CA, Magyar CI, Connolly HV, Fernandez ID, van Wijngaarden E. High prevalence of sleep disorders and associated comorbidities in a community sample of children with Down syndrome. J Clin Sleep Med. 2014;10:411-9; Churchill SS, Kieckhefer GM, Bjornson KF, Herting JR. Relationship between sleep disturbance and functional outcomes in daily life habits of children with Down syndrome. Sleep. 2015;38:61-71. 

Up to 97% of children with DS also have obstructive sleep apnea depending on the patient selection criteria, definitions and methodology used (Walter et al., Insights into the effects of sleep disordered breathing on the brain in infants and children: imaging and cerebral oxygenation measurements. Sleep Medicine Reviews. 2019 Dec 16;50:101251. doi: 10.1016/j.smrv.2019.101251). The review needs to discuss how sleep disorders in general and congenital heart disease may interact and in the papers identified if subjects had both sleep disorders and congenital heart disease.

No studies assessed the interaction between CHD and sleep disorders and their combined impact on cognitive function. This is clearly an area of unmet need, and our discussion has been revised to note this deficit in the literature (page 9, lines 171-175). We have also included the incidence of OSA across all included studies, when defined according to pediatric thresholds, in our discussion (page 9, lines 166-168) and have included reference to a recent meta-analysis determining the incidence of OSA in DS (70%). 

Thank you for the suggestion to include sleep behaviors. We have added a paragraph describing associations observed in these studies between cognition and the initiation/maintenance of sleep (page 7, lines 107-112).

2. Sleep disordered breathing (SDB) is the global term generally used to cover both central and obstructive breathing patterns during sleep. Obstructive SDB describes a spectrum of severity of disorders which ranges from primary snoring which is not associated with sleep disruption or gas exchange abnormalities to obstructive sleep apnoea (OSA). In children, OSA is usually defined as an obstructive apnoea hypopnoea index of > 1 event per hour of sleep. The authors need to rewrite the section of discussion which does not make it plain if the papers are talking about polysomnographically defined OSA or parental reports of SDB.

We sincerely appreciate this comment and note that SDB and OSA were not clearly defined in our review. We have clarified how the studies evaluated and defined SDB and OSA in the body of the text (page 6, lines 83-92), modified Table 1 accordingly, and also included a discussion of the standard definition employed in the pediatric setting and how adherence to this standard may have impacted the results of our review (page 9, 161-168).

3. The section on treatment of OSA also needs to be rewritten as adenotonsillectomy is the first line of treatment for OSA in both typically developing children and children with DS. Adenotonsillectomy is less effective in children with DS as the authors report, and CPAP is frequently used when OSA is not completely resolved with adenotonsillectomy. Other forms of surgery are also common in children with DS include tracheostomy, mandibular distraction osteogenesis for children with significant retrognathia, genioglossus advancement, rapid maxillary advancement, lingual tonsillectomy, tongue reduction, tongue hyoid advancement or suspension, uvulopalatopharyngoplasty, tonsillar pillar plication, and laryngotracheoplasty (Rosen D. [Rosen, 2011 #2752]. Current Opinion in Pulmonary Medicine. 2011;17:431-6) and recently hypoglossal nerve stimulation (Diercks GR, Wentland C, Keamy D, Kinane TB, Skotko B, de Guzman V, et al. Hypoglossal Nerve Stimulation in Adolescents With Down Syndrome and Obstructive Sleep Apnea. JAMA Otolaryngol Head Neck Surg. 2017).

The co-occurrence of sleep disorders and congenital heart disease should also be discussed regarding thyroid and seizure disorders.

The discussion has been re-written and re-organized to summarize current recommendations regarding the management of sleep disorders in DS. We have added the reference regarding surgical options as suggested. Older references have been removed, and primary intervention is now noted as adenotonsillectomy (page 10, lines 188-198). Regarding co-occurrence of conditions, please see our response to comment #1.

Reviewer 2

1.It is not clear why certain chronic health conditions were selected, e.g. why was hearing impairment (a common health condition) not included?

Thank you for this question. We selected the five most common chronic health conditions that occur in Down syndrome, and excluded neurosensory conditions as there is already data that presence of these comorbidities impact cognitive ability. We have clarified this in our introduction as follows, on page 3, lines 9-12). 

o ‘While the above-described health conditions are common in DS, to date, only neurosensory deficits have been assessed for a relationship with cognitive ability (17, 18). The objective of this systematic review is to address this gap in knowledge by determining from available evidence the impact of non-neurosensory chronic health conditions on cognitive functioning in DS.’

2. Was Mosaic Down syndrome included as it is not described as a search term.

Mosaic Down syndrome was not included as a search term, as it is often an exclusion criteria for studies evaluating outcomes in DS. Of the included studies, Breslin et al. and Rosser et al. specifically mention excluding subjects with mosaicism. 

3. Was hand searching and reference lists for articles carried out? if not, why was this not included as part of the methodology as it is standard practice. Please also state the date range within which articles were searched.

We relied on a medical librarian to conduct a complex search of the electronic database, and then supplemented this search with manual reference list scans. Specifically, articles referenced by the included studies underwent full text review to identify any additional publications not discovered using traditional indexing resources. A statement clarifying this aspect of our approach has been added to the Methods (page 5, line 63-65). The date range of studies searched has now been included in the Methods, page 4, lines 32-33.

4. What tool was used to assess the risk of bias? please provide a reference.

Thank you for this question. We used the Cochrane Risk of Bias in Systematic Reviews (ROBIS) tool to assess bias, and have now described this approach more completely in the Methods (page 5, lines 65-70) and in the Discussion (page 11, lines 217-220). The relevant reference has also been added. 

5.How was data extracted? was this done by one or more raters?

All potentially eligible articles were initially extracted by a medical librarian into a database. Both raters then independently reviewed the title and abstracts to determine initial eligibility (1st round), and then in the second round read the full articles to determine final eligibility. After each round, any conflicts were resolved by an independent third party. In the Methods, the review strategy has been revised to clarify how the articles were selected. 

6. No information is given on how the data was analysed. Given that quantitative outcomes were included, there should be an explicit statement of why a meta-analysis was not performed. There appears to be a narrative synthesis of data but there is no attempt to provide statistics for any of the relationships discussed.

Thank you for making this point. We have now included an explicit statement for why a meta-analysis was not performed, specifically, 1) because the studies included represent a wide range of chronic health conditions and employed inconsistent methods for assessment of cognitive ability, and 2) because we felt our question was sufficiently answered with qualitative data (page 11, lines 208-210). 

7. In the results section, it would be helpful to provide a summary of how many studies were of each type, the variation in terms of sample sizes and where the studies were conducted (this information should be included in the tables)

We agree, and have revised Table 1 to include this information, in addition to correcting a few errors we noted in the table. We have also provided information regarding the types of studies included in Table 1, as well as in the text (page 5, lines 74-77): 

o ‘The included articles comprised 7 prospective cohort studies, 4 cross-sectional studies, and 4 case-control studies that were predominantly conducted in the U.S. and U.K., as well as in Israel, Greece, Thailand, and Taiwan. Sample sizes varied from 12 to 226 (mean 64, median 38).’

8. In the discussion section, please provide a discussion of the limitations of the study (e.g. only three electronic databases searched, only English language articles etc...).

In response to this request, we have expanded our discussion of study limitations to include the search strategy that was employed (page 9, lines 206-207):

o ‘Limitations to this review include a search strategy that was restricted to two databases and only English language articles.’

---

## [Decision Letter · Decision Letter 1]

26 Aug 2020

PONE-D-20-14548R1

The Relationship between Chronic Health Conditions and Cognitive Deficits in Children, Adolescents, and Young Adults with Down Syndrome: A Systematic Review

PLOS ONE

Dear Dr. Gramatges,

Thank you for resubmitting your work to PLOS ONE. Please make the corrections posed by Reviewer #1 so I can render a decision on this manuscript.

**Comments to the Author**

1. If the authors have adequately addressed your comments raised in a previous round of review and you feel that this manuscript is now acceptable for publication, you may indicate that here to bypass the “Comments to the Author” section, enter your conflict of interest statement in the “Confidential to Editor” section, and submit your "Accept" recommendation.

Reviewer #1: (No Response)

Reviewer #2: All comments have been addressed

2. Is the manuscript technically sound, and do the data support the conclusions?

Reviewer #1: Yes

Reviewer #2: Yes

3. Has the statistical analysis been performed appropriately and rigorously? 

Reviewer #1: N/A

Reviewer #2: N/A

4. Have the authors made all data underlying the findings in their manuscript fully available?

Reviewer #1: Yes

Reviewer #2: Yes

5. Is the manuscript presented in an intelligible fashion and written in standard English?

Reviewer #1: Yes

Reviewer #2: Yes

6. Review Comments to the Author

Reviewer #1: The authors have addressed my comments, however the relationship between sleep disordered breathing and obstructive sleep apnoea is not quite correct. Sleep disordered breathing describes a spectrum of respiratory disorders during sleep which ranges from simple or primary snoring at the mild end which is not associated with sleep disruption or gas exchange abnormalities. Primary snoring is defined when children snore but there is < 1 apnoea or hypopnoea per hour of sleep. Obstructive sleep apnoea is associated with sleep disruption and gas exchange abnormalities and is usually defined when there is > 1 apnoea or hypopnoea per hour of sleep. Page 44 line 161 needs to be corrected apnea or hypopnea rather than and. Page 41 the section on SDB needs to be clarified. Sleep disorder should be changed to respiratory sleep disorder.

Definitions are available in Berry RB, Budhiraja R, Gottlieb DJ, et al. Rules for scoring respiratory events in sleep: Update of the 2007 AASM manual for the scoring of sleep and associated events. Deliberations of the sleep apnea definitions task force of the american academy of sleep medicine. J Clin Sleep Med 2012;8:597-619.

Reviewer #2: I am satisfied that the authors have addressed my comments and thank the authors for their response.

I have no further comments.

7. PLOS authors have the option to publish the peer review history of their article (what does this mean?). If published, this will include your full peer review and any attached files.

**Do you want your identity to be public for this peer review?** For information about this choice, including consent withdrawal, please see our Privacy Policy.

Reviewer #1: No

Reviewer #2: No

We look forward to receiving your revised manuscript.

Kind regards,

Stephen D. Ginsberg, Ph.D.

Section Editor

PLOS ONE

---

## [Author Response · Author response to Decision Letter 1]

27 Aug 2020

August 26, 2020

Editorial Board

PLOS One

Re: PONE-D-20-14548R1, revised manuscript, systematic review

Dear Editorial Board,

Thank you for reviewing our manuscript, entitled ‘The Relationship between Chronic Health Conditions and Cognitive Deficits in Children, Adolescents, and Young Adults with Down Syndrome: A Systematic Review.’ We have revised the manuscript, included with ‘tracked changes,’ and provide a point-by-point response to the reviewer comments detailed below. 

Thank you for your consideration of this systematic review for publication in the PLOS One.

Sincerely,

M. Monica Gramatges, MD, PhD

Associate Professor of Pediatrics

ORCID: 0000-0002-0947-104X

Review Comments to the Author

Reviewer 1

The authors have addressed my comments, however the relationship between sleep disordered breathing and obstructive sleep apnoea is not quite correct. Sleep disordered breathing describes a spectrum of respiratory disorders during sleep which ranges from simple or primary snoring at the mild end which is not associated with sleep disruption or gas exchange abnormalities. Primary snoring is defined when children snore but there is < 1 apnoea or hypopnoea per hour of sleep. Obstructive sleep apnoea is associated with sleep disruption and gas exchange abnormalities and is usually defined when there is > 1 apnoea or hypopnoea per hour of sleep. Page 44 line 161 needs to be corrected apnea or hypopnea rather than and. Page 41 the section on SDB needs to be clarified. Sleep disorder should be changed to respiratory sleep disorder.

Definitions are available in Berry RB, Budhiraja R, Gottlieb DJ, et al. Rules for scoring respiratory events in sleep: Update of the 2007 AASM manual for the scoring of sleep and associated events. Deliberations of the sleep apnea definitions task force of the american academy of sleep medicine. J Clin Sleep Med 2012;8:597-619.

Thank you for this request for clarification.

Line 161 has been corrected to read: 

“OSA has been defined in pediatric populations as at least 1 apnea or hypopnea event per hour of sleep.”

The recommended reference (Berry et al.) has been added as a citation here. 

Line 81 has been corrected to read:

“Seven of fifteen articles (47%) reported on the spectrum of respiratory sleep disorders observed in association with cognitive performance in DS.”

Similarly, the term ‘respiratory’ has been added as clarification in lines 83, 158, and 228.

---

## [Editor Report · Decision Letter 2]

31 Aug 2020

The Relationship between Chronic Health Conditions and Cognitive Deficits in Children, Adolescents, and Young Adults with Down Syndrome: A Systematic Review

PONE-D-20-14548R2

Dear Dr. Gramatges,

We’re pleased to inform you that your manuscript has been judged scientifically suitable for publication and will be formally accepted for publication once it meets all outstanding technical requirements.

Kind regards,

Stephen D. Ginsberg, Ph.D.

Section Editor

PLOS ONE

---

## [Editor Report · Acceptance letter]

2 Sep 2020

PONE-D-20-14548R2 

The Relationship between Chronic Health Conditions and Cognitive Deficits in Children, Adolescents, and Young Adults with Down Syndrome: A Systematic Review 

Dear Dr. Gramatges:

I'm pleased to inform you that your manuscript has been deemed suitable for publication in PLOS ONE. Congratulations! Your manuscript is now with our production department. 

Kind regards, 

on behalf of

Dr. Stephen D. Ginsberg 

Section Editor

PLOS ONE